# The Emerging Role of Non-Coding RNAs in the Regulation of Virus Replication and Resultant Cellular Pathologies

**DOI:** 10.3390/ijms23020815

**Published:** 2022-01-13

**Authors:** Soudeh Ghafouri-Fard, Bashdar Mahmud Hussen, Hazha Hadayat Jamal, Mohammad Taheri, Guive Sharifi

**Affiliations:** 1Department of Medical Genetics, School of Medicine, Shahid Beheshti University of Medical Sciences, Tehran P.O. Box 14155-6153, Iran; s.ghafourifard@sbmu.ac.ir; 2Department of Pharmacognosy, College of Pharmacy, Hawler Medical University, Erbil 44001, Iraq; Bashdar.Hussen@hmu.edu.krd; 3Center of Research and Strategic Studies, Lebanese French University, Erbil 44001, Iraq; 4Department of Biology, College of Education, Salahaddin University-Erbil, Erbil 44001, Iraq; hazha.hidayat@su.edu.krd; 5Institute of Human Genetics, Jena University Hospital, Am Klinikum 1, 07747 Jena, Germany; 6Skull Base Research Center, Loghman Hakim Hospital, Shahid Beheshti University of Medical Sciences, Tehran P.O. Box 14155-6153, Iran

**Keywords:** long non-coding RNA, miRNA, HSV, EBV, CMV

## Abstract

Non-coding RNAs, particularly lncRNAs and miRNAs, have recently been shown to regulate different steps in viral infections and induction of immune responses against viruses. Expressions of several host and viral lncRNAs have been found to be altered during viral infection. These lncRNAs can exert antiviral function via inhibition of viral infection or stimulation of antiviral immune response. Some other lncRNAs can promote viral replication or suppress antiviral responses. The current review summarizes the interaction between ncRNAs and herpes simplex virus, cytomegalovirus, and Epstein–Barr infections. The data presented in this review helps identify viral-related regulators and proposes novel strategies for the prevention and treatment of viral infection.

## 1. Introduction

The human genome consists of a variety of non-protein-coding DNA. A proportion of these genomic regions are transcribed into RNA. These non-coding RNAs (ncRNAs) are believed to have diverse roles in cellular functions. In addition to those with characterized functions, further functional ncRNAs certainly need to be discovered and categorized [1]. Since the first evidence of biological functions of transfer and ribosomal RNAs in the 1950s, several other classes of ncRNAs have been identified. Two classes of these transcripts, namely, long ncRNAs (lncRNAs) and microRNAs (miRNAs), have gained special attention because of their regulatory roles on gene expression. While lncRNAs regulate gene expression at different levels, miRNAs mainly act at a post-transcriptional level. LncRNAs and miRNAs have another distinctive feature arising from their size. While the former group is longer than 200 nt, the latter are approximately 22 nt in length. LncRNAs interplay with other RNA species, particularly miRNAs, in a way that they sequester them and decrease their bioavailability [2]. Besides this, lncRNAs have biological functions through serving as scaffolds and enhancer RNAs [3].

The regulatory role of miRNAs on gene expression is mediated through miRNAs pairing with the miRNA recognition elements in the mRNAs. These elements are found particularly in the 3’ untranslated region (UTR) of transcripts; however, they are also present in 5′UTRs and coding regions. When the RNA-induced silencing complex is recruited to miRNA recognition elements, the target mRNA is destabilized, or its expression is repressed [4]. Based on an individual miRNA’s ability to target numerous miRNA recognition elements and suppression of hundreds of mRNAs, it is estimated that more than 60% of human protein-coding genes can be targeted by miRNAs [2].

NcRNAs, particularly lncRNAs, have recently been shown to regulate different steps in viral infections and induce immune responses against viruses [5]. Expressions of several host and viral lncRNAs have been found to be altered during viral infection [2,6]. These lncRNAs can exert antiviral function via inhibition of viral infection or stimulation of antiviral immune response. Some other lncRNAs can promote viral replication or suppress antiviral responses [2].

In the current review, we summarize the interaction between ncRNAs and herpes simplex virus (HSV), cytomegalovirus (CMV), and Epstein–Barr (EBV) infections.

## 2. ncRNAs and HSV Infection

Kaposi’s sarcoma-associated HSV-encoded miRNAs have been shown to affect the expression of host lncRNAs such as Maternally Expressed 3 (MEG3), antisense non-coding RNA in the INK4 locus (ANRIL), and Urothelial Cancer Associated 1 (UCA1) in favor of cancer development. More than 120 host lncRNAs have been identified as putative targets for viral miRNAs. Notably, in addition to the miRNA-dependent route, this type of HSV can affect the expression of host lncRNAs through direct interactions between lncRNAs and latency-related proteins. The impact of HSV on UCA1 expression has pro-proliferative and pro-migratory effects on endothelial cells [7].

HSV-1 has also been shown to increase the expression of Nuclear Enriched Abundant Transcript 1 (NEAT1) and the establishment of paraspeckles through influencing Signal Transducer And Activator Of Transcription 3 (STAT3). NEAT1 and other paraspeckle constituents, namely, P54nrb and PSPC1, interact with HSV-1 genomic DNA. Paraspeckle Component 1 (PSPC1) binds with STAT3 to facilitate its recruitment to paraspeckles and increase its interplay with viral gene promoters. This interaction increases the expression of viral genes and the replication of viruses. Suppression of NEAT1 or STAT3 has improved the healing of HSV-1-related skin lesions in animal models [8]. Another study has shown that HSV infection can result in the construction of higher numbers of paraspeckles via increasing expression of NEAT1 [9]. NEAT1 has also been shown to cooperate with HEXIM1 to construct a multi-subunit complex that participates in the regulation of DNA-associated innate responses of the immune system. This complex encompasses DNA-PK subunits as well as paraspeckle proteins. In fact, binding of HEXIM1 to NEAT1 has an essential role in the assembly of this complex. This complex has a vital participation in induction of innate immune responses against foreign DNA via induction of cGAS-STING-IRF3 pathway [10].

A high throughput RNA sequencing experiment has shown over-expression of lncRNAs in murine 661 W cells following HSV-1 infection. U90926 RNA has been identified as the most over-expressed lncRNA after infection with this virus. Being located in the nucleus, U90926 enhanced replication of HSV-1 DNA and increased proliferation of this virus. U90926-silenced cells have exhibited higher survival rates [11].

During the latent phase of HSV infection, one region of its genome which encodes the latency-associated transcript (LAT) is not silenced. This lncRNA has been first identified as an enhancer of HSV-1 reactivation. Yet, subsequent studies have shown this lncRNA’s role in the inhibition of cell apoptosis and enhancement of the formation of latency. Experiments in a rabbit model have revealed that reduction in LAT levels in neurons after the establishment of the latent phase decreases the capacity of HSV to reactivate. Thus, the HSV-1 LAT transcript is involved in the reactivation in an independent manner from its role in establishing latency [12].

Kaposi’s sarcoma-associated HSV encodes a viral oncogene, namely, viral interferon regulatory factor 1 (vIRF1). This oncogene enhances migration potential and aggressiveness of endothelial cells through decreasing expression of miR-218-5p to release its targets HMGB2 and CMPK1 from its inhibitory effects. Functionally, vIRF1 suppresses the function of p53 to enhance the expression of DNA Methyltransferase 1 (DNMT1) and methylation of pre-miR-218-1 promoter (Figure 1). This process leads to the enhancement of the expression of OIP5 Antisense RNA 1 (OIP5-AS1), a lncRNA that sponges miR-218-5p. Cumulatively, the cellular lnc-OIP5-AS1/miR-218-5p axis is hijacked by vIRF1 to facilitate the invasiveness of HSV-associated tumors [13].

miR-H1 as an HSV-1-encoded miRNA has been shown to target Ubiquitin Protein Ligase E3 Component N-Recognin 1 (Ubr1) to increase amassing of neurodegeneration-associated protein [14]. Another study has demonstrated the role of miR-H1/H6 in the facilitation of effective reactivation from latency [15]. Moreover, HSV-1 has been shown to encode miR-H2-3p, a miRNA that interferes with cytosolic DNA-induced antiviral innate immune responses through decreasing expression of DEAD-Box Helicase 41 (DDX41) [16]. Table 1 and Table 2 show the role of host and viral-encoded ncRNAs in HSV infection.

## 3. ncRNAs and CMV Infection

RNA1.2 has been recognized as one of four principal lncRNAs which are expressed by human CMV. This lncRNA has an essential function in manipulating the cellular NF-κB-dependent pathways of cytokine and chemokine production in the course of CMV infection. Thus, this lncRNA can affect host immune responses [29]. Another study has shown that human CMV encodes lncRNA4.9, which is localized to the viral replication compartment in the nucleus. Depletion of this lncRNA decreases CMVV DNA replication and its growth. Notably, CRISPR-Cas9-mediated targeting the RNA4.9 promoter results in the reduction of viral ssDBP levels implying the relation between ssDBP levels and oriLyt activity [30].

An in vitro study has shown the ability of human CMV in infecting primary human mammary epithelial cells. This infection leads to inactivation of Rb and p53 proteins, enhancement of telomerase activity, and activation of c-Myc and Ras as well as Akt and STAT3 signaling pathways. CMV-transformed cells exhibited a CMV signature associated with the lncRNA4.9 gene. The sequence of this lncRNA has also been detected in xenograft tumors originated from CMV-transformed cells. Most notably, similar lncRNA4.9 genomic sequences have been found in tumor samples of breast cancer patients [31].

Persistent CMV infection in elderlies has been associated with down-regulation of Non-Coding Repressor Of NFAT (NRON) lncRNA, while up-regulation of its immunity-associated target gene NFAT, in both CD28nullCD8+ T cells and CMVpp65CD8+ T cells (Figure 1). Thus, NRON has been suggested to contribute to CMV-induced CD28nullCD8+ T cell aging through affecting IL-4-associated NFAT signals [32].

Experiments in animal models have shown that miR-1929-3p partakes in CMV-induced hypertensive vascular remodeling via inflammasome activation through Ednra/NOD-, LRR-, and pyrin domain-containing protein 3 (NLRP3) axis [33].

Another study has shown that human CMV encodes several ncRNAs such as miR-US5-1 and miR-UL112-3p that protect CD34+ hematopoietic progenitors from apoptosis through inactivation of Forkhead box class O 3a (FOXO3a) [34]. miR-UL112-3p has an established role in the enhancement of the progression of glioblastoma [35]. Table 3 and Table 4 show the role of host and viral-encoded ncRNAs in CMV infection.

## 4. ncRNAs and EBV Infection

Nasopharyngeal carcinoma is one of the EBV-associated cancers. EBV has been shown to express very few viral proteins in nasopharyngeal carcinoma cells, probably in order to evade induction of immune responses. Yet, it expresses high amounts of EBV BamHI-A region rightward transcript (BART) miRNAs and lncRNAs (Figure 1). These ncRNAs are implicated in the pathogenesis of EBV-related disorders. The expression of BARTs has been shown to be regulated by the NF-κB pathway. In fact, EBV LMP1 as an effective activator of the NF-κB pathway can increase the expression of BARTs via this pathway. Meanwhile, BART miRNAs can decrease the expression of LMP1. NF-κB pathway and expression of BARTs construct an autoregulatory circuit to preserve EBV latency in nasopharyngeal carcinoma cells [46].

BHLF1 gene of EBV has been shown to encode several lncRNAs in linear and circular forms that participate in viral replication. However, an open reading frame has been detected in this gene in a proportion of EBV isolates. BHLF1 transcripts have also been detected during the latent phase. In fact, this lncRNA participates in important features of EBV latency, such as its capacity to induce the constant proliferation of B lymphocytes and their malignant transformation [47]. On the other hand, the EBV-encoded miRNA miR-BART6-3p has been shown to act as a tumor suppressor. This miRNA could inhibit metastasis and invasion processes and suppress the proliferation of EBV-related neoplasms via decreasing the expression of LOC553103. LOC553103 has been found to directly bind with the 3’UTR of STMN1 and increase its stability. Cumulatively, miR-BART6-3p/LOC553103/STMN1 molecular route can modulate levels of cell cycle-related proteins, which subsequently suppress the EBV-related proliferation of tumor cells [48].

Over-expression of EBV-miR-BART5-3p has been shown to promote the growth of nasopharyngeal and gastric cancer cells. This miRNA can directly target 3’UTR of TP53 and subsequently decrease Cyclin Dependent Kinase Inhibitor 1A (CDKN1A), BCL2 Associated X, Apoptosis Regulator (BAX), and Fas Cell Surface Death Receptor expression levels. Thus, this miRNA accelerates cell cycle progression and inhibits cell apoptosis. Most notably, BART5-3p participates in chemo/radioresistance. Furthermore, it increases p53 protein degradation [49].

Huang et al. have sequenced more than 11,000 lncRNAs and 144,000 protein-coding transcripts from four EBV-associated and EBV-negative gastric cancer samples in addition to their adjacent unaffected tissues. They have shown specific expression of SNHG8 in EBV-associated gastric cancer. This lncRNA has been demonstrated to influence the activity of numerous gastric cancer-specific signaling pathways and genes being affected by EBV [50]. Table 5 and Table 6 show the role of host and viral-encoded ncRNAs in EBV infection.

## 5. Impact of Drugs on the Expression of ncRNAs in Infected Patients

EBV-miR-BART22 has been shown to promote stemness properties, metastatic abilities of cancer cells, and increase their resistance to cisplatin. This viral-encoded miRNA directly targets the MAP2K4 and up-regulates MYH9 levels through PI3K/AKT/c-Jun signaling. Notably, cinobufotalin has been found to suppress miR-BART22-associated cisplatin resistance through enhancing expression of MAP2K4 to inhibit MYH9/GSK3β/β-catenin cascade and EMT process in nasopharyngeal carcinoma [71]. Another study has shown that RAS^G12V^ and irinotecan up-regulate BART3-3p expression to hamper gastric cancer cells’ senescence and lower NK cells and macrophages infiltration. In fact, BART3-3p reduces TP53, TP21, and inflammatory cytokines such as IL-1A, IL-1B, IL-6, and IL-8 [72]. Table 7 shows the impact of drugs on the expression of ncRNAs in infected patients.

## 6. Diagnostic Value of Non-Coding RNAs in EBV-Infected Individuals

Expression levels of EBV-encoded miRNAs such as miR-BART2-5p, miR-BART7-3p, miR-BART9-3p, and miR-BART13-3p have been shown to distinguish patients with nasopharyngeal cancer from healthy controls with diagnostic values ranging from 0.87 to 0.97 (Table 8).

## 7. Discussion

The interactions between ncRNAs and viral genes result in different pathophysiological consequences, thus affecting the clinical course of infection. In the current review, we have focused on the role of lncRNAs and miRNAs in HSV, CMV, and EBV infections and the induction of antiviral response. Most of the conducted studies have summarized the expression pattern of host transcripts. However, virus-derived ncRNAs have also been studied.

A possible mechanism for the oncogenic function of HSV is hijacking some cellular elements such as lncRNAs and miRNAs. Interference with this process possibly has therapeutic effects. Moreover, interference with viral-encoded ncRNAs might affect pathogenic processes in the course of viral infections. For instance, LAT-targeting ribozymes have been suggested as a possible strategy for treating recurrent HSV-related diseases such as herpes stromal keratitis [12].

A number of viral-encoded lncRNAs and miRNAs regulate the establishment of permanent latency, which might be a preceding phase of tumor development. Since this phase might be continued for a long time, it provides a window for therapeutic interventions in order to reverse the carcinogenic process. These viral transcripts have been shown to affect the expression of both protein-coding genes and ncRNAs encoded by the host cells. Thus, the interactions between cellular and viral ncRNAs are entirely complicated, necessitating the conduction of high throughput sequencing experiments and integrative bioinformatics analyses.

A number of cellular signaling pathways, such as NF-κB signaling, can affect viral latency. Additional investigations about the mechanisms by which NF-κB signaling can regulate the latency of viral particles might reveal novel therapeutic options for the treatment of viral-associated cancers.

Since most viral-encoded miRNAs and lncRNAs do not have similar sequences in the human genome, they can be considered appropriate diagnostic biomarkers, particularly for viral-associated malignancies. This type of application has been studies for EBV-encoded miRNAs such as miR-BART2-5p, miR-BART7-3p, miR-BART9-3p, and miR-BART13-3p, showing promising results in EBV-associated nasopharyngeal carcinoma. Further research in other types of viral-associated cancers is needed to expand the diagnostic application of these ncRNAs.

The data presented in this review helps identify viral-related regulators and proposes novel strategies for the prevention and treatment of viral infection.

## Figures and Tables

**Figure 1 ijms-23-00815-f001:**
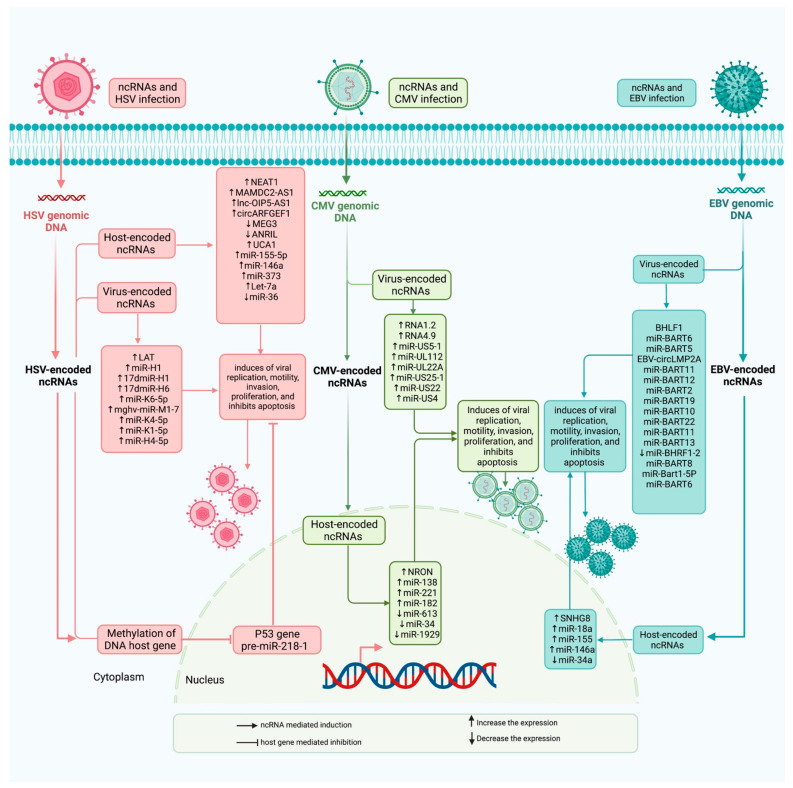
lncRNAs influence immune defense responses through directly interacting with host DNA genome, effecters, and transcriptional factors, and through releasing different types on non-coding RNAs, such as lncRNAs and miRNAs. Light pink represents HSV infection, light blue represents CMV infection, and pale turquoise represents EBV infection.

**Table 1 ijms-23-00815-t001:** Host-encoded ncRNAs and HSV infection (HEXIM1-DNA-PK-paraspeckle components-ribonucleoprotein complex (HDP-RNP), infected cell protein 0 (ICP0), thymidine kinase (TK), virion protein 16 (VP16), High-mobility-group protein 2 (HMGB2), SRSF2 (Serine and arginine Rich Splicing Factor 2), Interleukin-1 receptor-associated kinase (IRAK), RBPJ (Recombination Signal Binding Protein For Immunoglobulin Kappa J Region), interferon-induced transmembrane protein 1 (IFITM1)). (↑ upregulation; ↓ down regulation).

ncRNA	Expression Pattern	Clinical Samples/Animal Model	Cell Culture	Targets	Regulators	Signaling Pathways	Description	Reference
NEAT1	↑	-	HeLa, MEF	TK, ICP0	-	-	PSPC1 binds to STAT3 and increases STAT3 binding to the viral promoter, leading to increased viral expression. Knocking down STAT3 or NEAT1 improves skin healing.	[8]
	↑	-	HeLa, HeLa S3, HEK293T, HUVEC,	HDP-RNP, HEXIM1	-	-	NEAT1 assembles HDP-RNP, which regulates innate immune response against foreign DNA through stabilizing the host genome.	[10]
U90926	↑	-	661 W, Vero	ICP0, ICP4	-	-	After HSV-1 infection, upregulation of U90926 results in increased viral proliferation.	[11]
MAMDC2-AS1	↑	-	293T, A549, HaCaT, HepG2, HFF, HeLa	Hsp90α, VP16	YY1	-	This lncRNA increases HSV-1 infection in human cells through interacting with Hsp90α and importing HSV-1 into the nucleus.	[17]
MEG3	↓	-	TIVE, HUVEC, HeLa, iSLK, BCBL-1	-	miR-K12-3, miR-K12-5, miR-K12-6-5p, miR-K12-8, miR-K12-9	-	During the active infection and latency phase, HSV dysregulates lncRNAs expression using its miRNAs and proteins. Moreover, UCA1 induction promotes cell migration and proliferation.	[7]
ANRIL	↓			-	miR-K12-1, miR-K12-6-5p, miR-K12-2, miR-K12-11			
UCA1	↑			-	Kaposin and vCyclin			
lnc-OIP5-AS1	↑	-	iSLK, HEK293T, HUVECs	miR-218-5p, HMGB2, CMPK1	vIRF1	-	cIRF1 induces lnc-OIP5-AS1 upregulation, which diminishes miR-218-5p. As a result, in KSHV cells, migration and proliferation would be increased.	[13]
circARFGEF1	↑	-	iSLK-RGB-BAC16, iSLK-RGB-K9, HUVECs, HEK293T	miR-125a-3p, GLRX3	vIRF1	-	As a circRNA, ARFGEF1is induced by vIRF1. Moreover, ARFGEF1 increases cell motility, invasion, proliferation, and angiogenesis in Kaposi’s sarcoma-associated herpesvirus infected cells.	[18]
miR-155-5p	↑	-	HeLa	SRSF2	-	-	This miRNA induces improved levels of viral replication and gene expression by regulating SRSF2 promoter histones.	[19]
miR-146a	↑	-	THP-1, HEp-2	IRAK1	-	NF-κB	miR-146a activates NF-κB signaling pathway in HSV-1 infected cells.	[20]
let-7a	↑		293T, iSLK.219	RBPJ	LANA, LIN28B	NF-κB	In Kaposi’s sarcoma-associated herpesvirus infected cells, let-7a inhibits the lytic reactivated phase.	[21]
miR-36	↓	-	BJAB, HMVEC-d, HFFs, HEK293	IFITM1	-	-	miR-36 overexpression lowers IFITM1 levels, a protein induced by Kaposi’s sarcoma herpes virus, leading to decreased viral infection.	[22]
miR-373	↑	Serum samples from 10 herpetic gingivostomatitis and 10 normal cases	HeLa	IRF1	-	-	miR-373 is overexpressed upon HSV-1 infection, leading to increased viral replication and infection.	[23]

**Table 2 ijms-23-00815-t002:** Virus-encoded ncRNAs and HSV infection (DEAD-Box Helicase 41 (DDX41), EWSR1 (EWS RNA Binding Protein 1, Cytosolic arginine sensor for mTORC1 subunit 1 (CASTOR1), Suppressor Of Cytokine Signaling 2 (SOCS2)). (↑ upregulation; ↓ down regulation).

ncRNA	Expression Pattern	Clinical Samples/Animal Model	Cell Culture	Targets	Signaling Pathways	Description	Reference
LAT	↑	Rabbit skin cells	primary rabbit kidney cells	-	-	LAT knockdown diminishes HSV-1 reactivation in the latency phase. Therefore, this lncRNA could be used as a therapeutic target for herpes stromal keratitis.	[12]
	↑	Human fetal skin tissue, SCID C.B-17 male mice	-	-	-	LAT is a remarkable factor in increasing HSV-1 replication and lesion formation.	[24]
miR-H1	↑	-	SH-SY5Y, HEK293T		-	miR-H1, encoded by HSV-1, lowers Ubr1, a ubiquitin-protein ligase, which leads to β-amyloid accumulation.	[14]
17dmiR-H1	↑	female ND-40 Swiss-Webster mice, Male and female New Zealand White rabbits	CCL-13, HEK293T	LAT	-	Upon deleting these miRNAs, HSV-1 loses the ability to reactivate.	[15]
17dmiR-H6	↑						
miR-K6-5p	↑	-	HEK293T, BC-3, BC-1, JSC-1, BCBL-1	CCND3, CDC25A	-	In Kaposi’s sarcoma-associated herpesvirus, this miRNA is expressed and as a tumor suppressor lowers cell cycle progression.	[25]
miR-H2-3p	↑	-	HEK293T, THP-1, HFF	DDX41, IFN-β, MxI	-	miR-H2-3p lowers innate immune response in infected cells and intensifies viral DNA replication and proliferation.	[16]
mghv-miR-M1-7-5p	↑	C57BL/6J mice	NIH3T12	EWSR1	-	This miRNA is encoded by gamma-herpesviruses and is a necessary factor for its infection and splenic latency.	[21]
miR-K4-5p	↑	-	TIVE, KTIVE, MM, KMM	CASTOR1	mTORC1	These two miRNAs are encoded by Kaposi’s sarcoma herpes virus and could activate the mTORC1 signaling pathway and improve cell proliferation and colony formation.	[26]
miR-K1-5p	↑						
miR-H4-5p	↑	-	HeLa	CDKN2A, CDKL2	-	miR-H4-5p is transcribed by HSV-2 and induces cell cycle progression, proliferation, and inhibits apoptosis.	[27]
miR-H9-5p	↑	10 cancerous lung tissues and 10 control tissues	LTEP-α-2, SPC-α-1	SOCS2		Transcribed by HSV-2 infection, miR-H9-5p increases migration, proliferation, and invasion of lung cancer cells.	[28]

**Table 3 ijms-23-00815-t003:** Host-encoded ncRNAs and CMV infection (Endothelin-1 (ET-1), Sirtuin 1 (SIRT1), NLRP3 (NOD-, LRR-, and pyrin domain-containing protein 3), FOXO3a (Forkhead box class O 3a)). (↑ upregulation; ↓ down regulation).

ncRNA	Expression Pattern	Clinical Samples/Animal Model	Cell Culture	Targets	Signaling Pathways	Description	Reference
NRON	↑	40 elderly CMV positive cases	-	NFAT	-	Early CMV infection results in increased NRON expression in all B cell types.	[32]
miR-1929-3p	↓	C57BL/6 J mice	-	ET-1, Ednra, NLRP3	-	CMV downregulates miR-1929-3p to enhance blood pressure, endothelial cell injury, and vascular remodeling in mice.	[33]
miR-138	↑	-	HUVECs, MRC-5	SIRT1, p-STAT3	-	Endothelial cells infected by CMV indicate a higher miR-138 expression, which leads to tube formation and migration.	[36]
	↑	Male BALB/c nude mice	MNK-45, SGC-7901	IL6R	-	UL136 induces cell invasion and proliferation by activating IL6/STAT3 signaling, causing fluctuation in these miRNAs’ expression.	[37]
miR-34	↓						
miR-221	↑	C57BL/6 mice	Neural Precursor Cells	SOCS1, IFN	NF-κB	miR-221 hampers CMV replication in cells through modulating the NF-κB signaling pathway.	[38]
miR-182	↑	Female Balb/c mice	U-251MG, HFFs, NPCs	IRF7, FOXO3, IFN-I	-	CMV infection increases miR-182 expression, which itself produces IFN-I to restrict CMV replication.	[39]
miR-613	↓	10 CMV-positive and 10 CMV-negative glioblastoma tissues and their adjacent normal tissues	U87, U251	Arginase-2	-	In glioblastoma infected cells and tissues, miR-613 is negatively correlated with tumor size, stage, and patients’ survival rates. Moreover, it could diminish colony formation, migration, and invasion.	[40]

**Table 4 ijms-23-00815-t004:** Virus-encoded ncRNAs and CMV infection (Tumor protein p63-regulated gene 1-like protein (TPRG1L), Monocyte Chemoattractant Protein-1 (MCP-1), CXCL1 (C-X-C Motif Chemokine Ligand 1, BCL2L11 (BCL2 Like 11), TUSC3 (Tumor Suppressor Candidate 3), GAB1 (GRB2 Associated Binding Protein 1)). (↑ upregulation; ↓ down regulation).

ncRNA	Expression Pattern	Clinical Samples/Animal Model	Cell Culture	Targets	Signaling Pathways	Description	Reference
RNA1.2	↑	-	HFFF2, HEK293T, HFT	TPRG1L, IL-6, MCP-1, CXCL1	NF-κB	RNA1.2, encoded by CMV, modifies the expression of many genes, which is a conspicuous indication of modulating the NF-κB signaling pathway.	[29]
RNA4.9	↑	-	MEF	ssDBP	-	Located in the nucleus, RNA4.9 improves viral growth and DNA replication.	[30]
	↑	female NOD/SCID Gamma mice	MDA-MB-231, MCF-7, HMECs	-	-	CMV infection induces RNA4.9 expression, which leads to tumor formation.	[31]
miR-US5-1	↑	-	CD34+ HPCs of fetal liver tissues, THF, NHDF, HEK293	FOXO3a, BCL2L11	MAPK	CMV downregulates pro-apoptotic proteins and increases CD34^+^ progenitor cells.	[34]
miR-UL112-3p	↑						
	↑	40 CMV-positive glioblastoma and adjacent normal tissues	Glioblastoma primary culture, HEK293	TUSC3	Akt	miR-UL112-3p improves glioblastoma cells proliferation, migration, invasion, and colony formation.	[35]
miR-US5-2	↓	-	NHDF, hAECs, 293T	GAB1, UL138, EGR1	PI3k/ERK	This miRNA can thwart the normal EGF signaling pathway that boosts cell proliferation.	[41]
	↑	-	NHDF, HEK293T, CC-2535	TGF-β, NAB1	-	HCMV miR-US5-2 is a latent miRNA that suppresses uninfected CD34^+^ hematopoietic progenitor cells.	[42]
miR-UL22A	↑			SMAD3		By diminishing SMAD3, this miRNA reactivates CMV in CD34^+^ cells.	
miR-US22	↑	-	HEK293T, NHDF, AEC	EGR1	-	miR-US22 lowers hematopoietic stem cells proliferation, self-renewal, and colony formation.	[43]
miR-US25-1-5p	↑	-	HFF, U251, HEK293	CD147, IFN-β	NF-κB	miR-US25-1-5p diminishes antiviral immune response and CD147 but induces the lytic phase of CMV virus.	[44]
miR-US4-5p	↑	-	HELF, HEK293, THP-1	PAK2	-	miR-US4-5p, encoded by CMV, inhibits apoptosis by downregulating PAK2.	[45]

**Table 5 ijms-23-00815-t005:** Host-encoded ncRNAs and EBV infection (TRIM28 (Tripartite Motif Containing 28), EIF4A2 (Eukaryotic Translation Initiation Factor 4A2), NAP1L1 (Nucleosome Assembly Protein 1 Like 1), PLD3 (Phospholipase D Family Member 3), RPL18A (Ribosomal Protein L18a), SMAD2 (SMAD Family Member 2), CXCR4 (C-X-C Motif Chemokine Receptor 4), Programmed death-ligand 1 (PD-L1)). (↑ upregulation; ↓ down regulation).

ncRNA	Expression Pattern	Clinical Samples/Animal Model	Cell Culture	Targets	Regulators	Signaling Pathways	Description	Reference
SNHG8	↑	88 gastric cancer and adjacent normal tissues	-	TRIM28, EIF4A2, NAP1L1, PLD3, RPL18A, TRPM7		-	SNHG8 is up-regulated in gastric cancer tissues afflicted with the EBV virus and may trigger cancer initiation by altering various proteins.	[50]
miR-18a	↑	21 cancerous and 14 non-cancerous tissues, male BALB/c nude mice	CNE1, CNE2, S-18, S-26, 5-8F, 6-10B, SUNE2, C666-1	SMG1	LAT	NF-κB	In EBV-positive cases, miR-182 positively correlates with nasopharyngeal tumor size and tumor stage. Moreover, it enhances cell migration, invasion, and proliferation.	[51]
miR-155-5p	↑	-	GT38, GT39, SNU719, HGC27, SGC7901, BGC823	Smad2	LMP2A	NF-κB, TGF-β	In EBV-infected gastric cancer cells, miR-155-5p reduces cell proliferation and boosts cell cycle arrest and apoptosis.	[52]
miR-146a	↑	-	GT38, GT39, AGS, BGC823, SGC7901	CXCR4	LMP1	-	EBV induces miR-146a expression, which leads to lower cell proliferation, migration, and cell cycle progression.	[53]
miR-34a	↓	27 diffuse large B-cell lymphoma cases	LCL, OMA4, DG75, BL41, U2932, SUDHL5, ER/EB 2.5, Mutu I, Mutu III, Daudi, Jijoye	PD-L1	EBF1	-	In Burkitt lymphoma and diffuse large B-cell lymphomas, miR-34a acts as a tumor suppressor due to downregulating PD-L1 and activating T cells.	[54]
	↓	-	SNU719, SNU638	NOX2	EBNA1	-	In gastric cancer cell lines, miR-34a downregulation leads to increased cell viability and reduced apoptosis rate.	[55]

**Table 6 ijms-23-00815-t006:** Virus-encoded ncRNAs and EBV infection (AT-rich interactive domain-containing protein 1A (ARID1A), latent membrane protein 1 (LMP1), Tubulin polymerization promoting protein 1 (Tppp1), DKK1 (Dickkopf WNT Signaling Pathway Inhibitor 1), NKIRAS2 (NFKB Inhibitor Interacting Ras Like 2)). (↑ upregulation; ↓ down regulation).

ncRNA	Expression Pattern	Clinical Samples/Animal Model	Cell Culture	Targets	Signaling Pathways	Description	Reference
BHLF1	↑	-	A.21, Kem I, Mutu I, BX1, HEK293	-	-	After being up-regulated by SM protein, BHLF1 encodes ncRNAs that induce a latency phase, immortalize EBV, and generate B cells growth.	[47]
miR-BART6-3p	↑	female BALB/c nude mice	5-8 F, HNE2, C666-1, HEK293T	LOC553103, STMN1	-	Invasion, cell cycle progression, and metastasis of tumors are all diminished by this miRNA.	[48]
miR-BART5-5p	↑	-	CNE2, HeLa-Bx1, HEK293T, C666-1	LMP1	NF-κB	Through the NF-κB pathway, LMP1 activates the BART promoters in nasopharyngeal carcinoma. BART itself could downregulate LMP1 expression as well.	[46]
	↑	24 gastric cancer patients, nude mice	SGC7901, GES1, HNE1, 6-10B, AGS	TP53, CDKN1A, BAX, FAS	-	Aside from increasing the resistance to chemotherapeutic agents, miR-BART5-5p improves cell cycle progression and growth in cancerous tissues.	[49]
EBV-circLMP2A	↑	78 gastric cancer patients, female NOD/SCID mice	SNU719, YCCEL1, HEK293T	miR-3908, TRIM59, p53	-	This circRNA is encoded by EBV and has a remarkable association with poor prognosis and metastasis in EBV-associated gastric cancer cases. Furthermore, it induces stemness in cancerous cells.	[56]
miR-BART11-3p	↑	20 EBV positive and 2 EBV negative cases	MKN7, NCI-N87	ARID1A	-	These two miRNAs bind ARID1A, a tumor suppressor, and decrease its level in gastric cancer tumors.	[57]
miR-BART12	↑						
	↑	27 cancerous and 13 nasopharyngeal epithelial tissues, male BALB/c nude mice	C666-1, 5-8F, AGS	TPPP1	-	This miRNA increases the invasion, EMT, and migration rate in EBV-related tumors by hampering α-tubulin acetylation and remodeling the cytoskeleton. It is also correlated with a poor prognosis rate.	[58]
	↑	55 nasopharyngeal tumors and 21 normal tissues	C666-1, NPC43, C17, HK1, NP69, C15, HeLa, 293FT	BRCA1	-	These miRNAs downregulate BRCA1 and potentiate cancerous cells to chemotherapy.	[59]
miR-BART2-3p	↑						
miR-BART17-5p	↑						
miR-BART19-3p	↑						
	↑	20 chronic active EBV infection, 20 EBV-associated hemophagocytic lymphohistiocytosis, 10 healthy cases	AGS, C666-1, B95-8, Akata-Bx1	APC	-	miR-BART19-3p is increased in these diseases and induces cell proliferation, and inhibits cell apoptosis.	[60]
miR-BART10-3p	↑	-	AGS, SNU-719, HEK293T	DKK1	-	In gastric cancer cells infected by EBV, this miRNA increases cell proliferation and migration.	[61]
	↑	874 gastric cancerous tissues	SNU719, YCCEL1, BGC823, AGS	APC, DKK1, Twist	Wnt	These miRNAs induce invasion and migration and are indicators of a lower survival rate.	[62]
miR-BART22	↑						
miR-BART11	↑	36 gastric cancer tissues, blood samples from 102 gastric patients, and 112 healthy controls	AGS, THP-1, MKN-45, SGC-7901, 293T	FOXP1	-	EMT is significantly increased by this miRNA. Moreover, it negatively correlates with a lower survival rate.	[63]
miR-BART1-5p	↑	28 EBV-positive and 31 EBV-negative gastric cancer patients.	GT38, GT39, SNU719, AGS, HGC27, BGC823, SGC7901, HEK293T	GCNT3	NF-kB	miR-BART1-5p inhibits cell migration and invasion in EBV-infected gastric cancer cells.	[64]
miR-BART13-3p	↑	24 cancerous tissues, BALB/c nude mice	CNE1, S26, CNE2, 5-8F, HEK293T	ABI2	c-JUN/SLUG	This miRNA could induce EMT, migration, and invasion in nasopharyngeal carcinoma through the c-JUN/SLUG pathway.	[65]
miR-BHRF1-2-5p	↓	-	PBMCs from healthy cases	PD-L1, PD-L2, LMP1	-	In progenitor B cells, miR-BHRF1-2-5p manipulates apoptosis-related molecules, insinuating its significant role in B cell lymphomas.	[66]
	↓	-	BJAB, HEK293T, isolated primary B lymphocytes	IL1R1, TYW3	NF-κB	miR-BHRF1-2-5p blocks NF-κB activation and IL-1 signaling.	[44]
miR-BART8-3p	↑	male nude mice	HONE1, 5-8F	γ-H2AX, CCNB1, CDK1, CHK1, CHK2	ATM/ATR	miR-BART8-3p, encoded by EBV, enhances tumor size, cell proliferation, and resistance to radiation therapy by activating the ATM/ATR pathway.	[53]
	↑	19 cancerous and 10 normal nasopharyngeal tissues, female BALB/c nude mice	CNE-1, SUNE-1, HEK293T, C666–1	RNF38	NF-κB, Erk1/2	miR-BART8-3p escalates migration, EMT, invasion, and metastasis in nasopharyngeal cancerous cells.	[67]
miR-BART13	↑	36 nasopharyngeal cancerous and 25 normal tissues, female BALB/c nude mice	CNE-1, 293T, C666-1, SUNE-1	NKIRAS2	NF-κB	miR-BART13 improves EMT, metastasis, tumor growth, and cell proliferation in cancerous cells by activating the NF-κB signaling pathway.	[68]
miR-Bart1-5P	↑	55 cancerous and 15 normal nasopharyngeal tissues, nude mice	HONE1, HK1, CNE1, 5-8F, 6-10B, SUNE1, HNE1 and CNE2, HEK293T	AMPKα1	AMPK/mTOR	miR-Bart1-5P augments glycosis, proliferation, and angiogenesis levels in nasopharyngeal carcinoma cells.	[69]
miR-BART6-3p	↑	-	HK-1, C666-1, BJAB, B95.8	IFN-β, RIG-I		In EBV infected cells, miR-BART6-3p impedes immune response and by downregulating a pattern recognition receptor.	[70]

**Table 7 ijms-23-00815-t007:** Impact of drugs on the expression of ncRNAs in infected patients. (↑ upregulation; ↓ down regulation).

ncRNA	Expression Pattern	Drug	Assessed Sample	Description	Reference
miR-BART22	↓	Cinobufotalin	HONE1 and 5-8F cells, 61 cancerous and 36 non-cancerous nasopharyngeal tissues, female BALB/c nude mice	Cinobufotalin downregulates miR-BART22, an ncRNA that enhances metastasis and tumor stemness in EBV-infected cases suffering from nasopharyngeal cancer. This miRNA increases cisplatin resistance by increasing MYH9 levels and activating PI3K/AKT/c-Jun pathway.	[71]
miR-BART3-3p	↑	RAS^G12V^/irinotecan	SGC7901, KATOIII, AGS, and HEK293 cells lines. 20 positive and 20 negative gastric cancer tissues + nude mice	These two drugs up-regulate BART3-3p expression to hamper gastric cancer cells’ senescence and lower NK cells and macrophages infiltration. That is to say, BART3-3p reduces TP53, TP21, and inflammatory cytokines such as IL-1A, IL-1B, IL-6, and IL-8.	[72]

**Table 8 ijms-23-00815-t008:** Diagnostic Value of ncRNAs in EBV-infected individuals.

ncRNA	Clinical Cases	AUC	Sensitivity	Specificity	Description	Reference
miR-BART7-3p	483 cases with nasopharyngeal carcinoma and 243 healthy cases	0.964	96.1	96.7	Advanced stage nasopharyngeal carcinoma is markedly correlated with these miRNAs overexpression. Therefore, they have the potential to be used as biomarkers for predicting patients’ outcomes.	[73]
miR-BART13-3p		0.973	97.3	99.6		
miR-BART13-3p	Serum samples from 39 nasopharyngeal carcinoma cases, 33 healthy controls, and 29 non-nasopharyngeal cases	0.91	74	97	Combining BART13-3p with BART7-3p results in an AUC equal to 0.93, indicating the significance of these miRNAs in diagnosing nasopharyngeal carcinoma.	[74]
miR- BART7-3p		0.90	85	90		
miR- BART9-3p		0.87	97	82		
miR-BART2-5p	148 nasopharyngeal cases and 118 healthy controls	0.972	93.9	89.8	-	[75]

## Data Availability

Not applicable.

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
