# Peer review of "The Emerging Role of Non-Coding RNAs in the Regulation of Virus Replication and Resultant Cellular Pathologies"

_ijms, 2022, doi:10.3390/ijms23020815_

Round 1
Reviewer 1 Report
The role of non-coding RNAs is introduced, especially in herpes 56 simplex virus (HSV), cytomegalovirus (CMV), and Epstein-Barr (EBV) infections. This review is of great interest to the readers of this journal because non-coding RNAs are one of the most important signal transductions for understanding the signal transduction system, which is not understood only from the genome. Therefore, I recommend this review after the following items be revised for publication.
#1: References on pages 15-17 are duplicated. Correct it.
#2: Explain the abbreviations as much as possible. For example, UTR: line 44), the name of non-coding RNA (NEAT1, MEG3 and so on)
#3: The Targets / Regulators column in Tables is confusing. In particular, it is difficult to understand when it is described as a set and when it is not, as in the column of [7]. Some modifications for this column are required.
Author Response
- We corrected the mentioned point about references.
- We explained the abbreviations.
- We modified this column.
Reviewer 2 Report
The review tackles an important issue, i.e. the role of non-coding RNAs in virus infections, focusing in particular on herpes simplex, cytomegalovirus and Epstein-Barr infections.
However the entire review needs a profound rethinking, since it needs to be more critical in the presentation of the various papers. In some of the cases the effect of the miRNAs or of the lncRNAs is not clear (for example lines 74-76). Moreover, in a lot of cases the target proteins are identified just with the acronyms, with no explanation of the role of that specific protein within the cell.
The tables, although comprehensive, are quite difficult to read, also because of the extreme number of abbreviations included (at least a legend should be added). It would be better if the authors were able to summarize at least part of the findings in figures.
Author Response
we applied all the comments according to the reviewer suggestion.
Round 2
Reviewer 2 Report
The manuscript has been improved